# The Effects of Concurrent Training Combined with Low-Carbohydrate High-Fat Ketogenic Diet on Body Composition and Aerobic Performance: A Systematic Review and Meta-Analysis

**DOI:** 10.3390/ijerph191811542

**Published:** 2022-09-14

**Authors:** Yubo Wang, Kaixiang Zhou, Vienna Wang, Dapeng Bao, Junhong Zhou

**Affiliations:** 1China Institute of Sport and Health Science, Beijing Sport University, Beijing 100084, China; 2Sports Health College, Chengdu University of Traditional Chinese Medicine, Chengdu 611137, China; 3College of Engineering, California State University, 1250 Bellflower Boulevard, Long Beach, CA 90840, USA; 4Hebrew Senior Life Hinda and Arthur Marcus Institute for Aging Research, Harvard Medical School, Boston, MA 02115, USA

**Keywords:** low-carb, high-fat ketogenic diet, concurrent training, body composition, aerobic performance, athletes

## Abstract

(1) Background: Recently, studies have emerged to explore the effects of concurrent training (CT) with a low-carb, high-fat ketogenic diet (LCHF) on body composition and aerobic performance and observed its benefits. However, a large variance in the study design and observations is presented, which needs to be comprehensively assessed. We here thus completed a systematic review and meta-analysis to characterize the effects of the intervention combining CT and LCHF on body composition and aerobic capacity in people with training experience as compared to that combining CT and other dietary strategies. (2) Methods: A search strategy based on the PICOS principle was used to find literature in the databases of PubMed, Web of Science, EBSCO, Sport-discuss, and Medline. The quality and risk of bias in the studies were independently assessed by two researchers. (3) Result: Eight studies consisting of 170 participants were included in this work. The pooled results showed no significant effects of CT with LCHF on lean mass (SMD = −0.08, 95% CI −0.44 to 0.3, *p* = 0.69), body fat percentage (SMD = −0.29, 95% CI −0.66 to 0.08, *p* = 0.13), body mass (SMD = −0.21, 95% CI −0.53 to 0.11, *p* = 0.2), VO_2max_ (SMD = −0.01, 95% CI −0.4 to 0.37, *p* = 0.95), and time (or distance) to complete the aerobic tests (SMD = −0.02, 95% CI −0.41 to 0.37, *p* = 0.1). Subgroup analyses also showed that the training background of participants (i.e., recreationally trained participants or professionally trained participants) and intervention duration (e.g., > or ≤six weeks) did not significantly affect the results. (4) Conclusions: This systematic review and meta-analysis provide evidence that compared to other dietary strategies, using LCHF with CT cannot induce greater benefits for lean mass, body fat percentage, body mass, VO_2max_, and aerobic performance in trained participants.

## 1. Introduction

The appropriate design of an exercise program by combining physical exercise with a daily diet is believed to be critical to improving body composition (e.g., lean body mass) and increasing functions (e.g., aerobic capacity) in trained participants, including athletes [1,2,3,4]. Concurrent Training (CT), a training method that combines endurance exercise and resistance training, has been evidenced to be a better strategy for improving aerobic capacity as compared to using solely endurance training [5,6,7,8,9]. Several studies have further shown that an appropriate diet can augment the effects of CT on aerobic capacity and body composition in athletes [4,10,11,12]. For example, CT combined with a high-carbohydrate diet can increase muscle glycogen stores and thus benefit aerobic capacity in trained participants (e.g., with experience of at least two-month regular training/exercise) [13]. However, several other studies suggested that the endurance capacity may not benefit from a high-carb diet due to limited glycogen stores, especially since it also increases the risk of obesity [14,15,16,17]. Therefore, it is critical to implement appropriate dietary strategies during the CT protocol.

Growing evidence suggests that reducing carbohydrate intake may increase systemic lipid oxidation and improve aerobic capacity in trained participants [18,19,20]. Recently, studies have emerged to explore the effects of a low-carb, high-fat ketogenic diet (LCHF) on aerobic capacity and body composition in trained participants [16,21,22]. LCHF is a dietary structure that reduces carbohydrate availability which can decrease carbohydrate intake to a maximum of approximately 50 g/d, or 5% of daily energy intake, and have protein consumption at a moderate or high level (e.g., 1.2 to 1.5 g/kg/d), leading to the predominant energy intake from fat [23]. Studies have shown that the use of LCHF improved fat utilization in trained participants during long-distance aerobic capacity [16,24,25,26,27]. For example, McSwiney et al., showed that 12-week CT combined with LCHF improved lean body mass in athletes of running [26]. However, several studies suggested that fat is an inferior metabolic fuel that cannot support higher-intensity exercise [28,29,30]; therefore, the fat-dominated strategy from LCHF may not be optimal. This is also supported by the findings from several studies [31,32,33]. For example, Zinn et al. observed a decrease in aerobic performance and no improvement in lean body mass in trained participants following a 10-week training of CT combined with LCHF [33]. To date, meta-analyses only focused on the effect of endurance training, not CT, combined with LCHF on aerobic performance in trained participants [34,35]. The effects of CT combined with LCHF on body composition and aerobic performance have not been examined, and the influences of the design of intervention protocols (e.g., training modality) on its effects are unclear.

To highlight the most recent study findings in this field and to explicitly examine the effects of CT combined with LCHF on body composition (e.g., lean mass, fat) and aerobic capacity in participants, we conducted a systematic review and performed a meta-analysis and sub-group analyses based upon peer-reviewed publications. This work may provide critical knowledge of the appropriate intervention design, ultimately informing future studies with rigorous design.

## 2. Materials and Methods

This systematic review and meta-analysis was conducted using preferred reporting items for systematic reviews and meta-analysis guidelines [36] and registered with PROSPERO (Registration ID: CRD42022310629), an international prospective registry for systematic reviews.

### 2.1. Literature Search

Two authors (YB and KZ) independently searched PubMed, EMBASE, EBSCO, and Web of Science from 1 January 2011 to 10 May 2022 using a comprehensive search strategy (Appendix A). A secondary search strategy was also used which involved a manual search of the reference lists of selected articles. Searches were limited to the English language only. Systematic review and meta-analysis of the data were performed in STATA16.0 software according to the Preferred Reporting Items of Systematic Reviews and Meta-Analysis (PRISMA) guidelines [37]. 

### 2.2. Selection Criteria

Articles were included if they met the following criteria:the participants were adults over 18 years of age without the disease (e.g., the presence of cardiovascular diseases, diabetes mellitus, arterial hypertension, or any other diseases that required pharmacological treatment);the intervention used in the study was CT with LCHF;the primary outcomes are lean mass (body composition) and VO_2max_ (aerobic performance);the secondary outcome are body mass, body fat percentage, and time (or distance) to complete the aerobic tests (100 km total time, 10 km total time, Yo-Yo total distance, total fatigue time and, total performance time);the design of the study was a randomized (or not randomized) controlled trial;the intervention duration was more than two weeks.

Articles were excluded if the language was not English, animal models were used, or the data were not extractable. Reviews and conference articles were also excluded from the analysis.

### 2.3. Data Extraction

The process of data extraction was conducted according to the Cochrane Collaboration Handbook [36]. Two authors (YW and KZ) independently performed data extraction, and when a decision disagreement happened, it was discussed with the third author (JZ) until a consensus was achieved. The extracted information from the publications included: the study (authors, year), participants (age, sex, training experience), grouping and sample size, interventions (diet type, training type, frequency, intervention duration), and outcome measures. 

Most studies only report data for pre- and post-intervention. Thus, average change was calculated as the difference between the mean of data pre- and post-intervention, and standard deviation (SD) was calculated using the following formula based on the principles of the Cochrane Handbook for Systematic Reviews of Interventions [36,38].
$$\mathrm{SDchange}=\sqrt{{\mathrm{SD}}_{\mathrm{pre}}^{2}+{\mathrm{SD}}_{\mathrm{post}}^{2}-2\times \mathrm{Corr}\times {\mathrm{SD}}_{\mathrm{pre}}\times {\mathrm{SD}}_{\mathrm{post}}}$$

If any relevant data were missing, we tried to contact the corresponding author or other authors of that study via email to request data [36].

### 2.4. Quality Assessment

The quality of the included studies was assessed independently by two authors (Y.W. and K.Z.) based on the principles of the Physiotherapy Evidence Database (PEDro). The PEDro scale includes 11 items, and each study was assessed as either “yes” (score 1) or “no” (score 0) for each of those items. According to the PEDro guidelines, the maximum total score was 10 (Item 1 is not used to compute the total score). If a study received a score of 9 or 10, it was considered to be of very good quality; a score of 6 to 8 reflected good quality; a score of 4 or 5 showed medium quality and a score of 0 to 3 suggested poor quality [39]. Any score on which the two authors disagreed was discussed with a third author (J.Z.) until a consensus was achieved.

### 2.5. Statistical Analysis

To determine the effect size of the intervention, the standardized mean differences (SMDs) of the outcomes were calculated (i.e., the mean difference between the effect of LCHF and control divided by the pooled standard deviation). Effect sizes were classified as trivial (<0.2), small (0.2~0.5), moderate (0.5~0.8), or large (>0.8) [40]. Meta-analysis was performed in Stata v16.0 (STATA Corp., College Station, TX, USA) using the inverse variance method for included studies that compared the effects of LCHF and control conditions on each included outcome. The statistical heterogeneity was evaluated using heterogeneity Chi-squared (χ^2^) and I^2^ values. The level of heterogeneity was interpreted according to guidelines from the Cochrane Collaboration: I^2^ values of 25%, 50%, and 75% for low, moderate, and high heterogeneity, respectively [41]. Due to the homogeneity of the included studies, in combination with a relatively low number of studies with small sample sizes, fixed-effect models were used. In accordance with the Cochrane recommendation [42]. However, in case of indications of heterogeneity (*p* < 0.1 and I^2^ > 50%), additional analyses were conducted using a random-effect model. In addition, publication bias was assessed by generating funnel plots and conducting Egger’s test. If a significant asymmetry was detected, we estimated the magnitude of the small study effect using the Trim and Fill method [43].

In addition to examining the place where heterogeneity may exist, we performed a series of subgroup analyses to examine the influences of different characteristics in the studies on the observations using the fixed-effect model [42]. Specifically, the analyses were performed based upon the training background of participants (i.e., recreationally or professionally trained participants) and intervention duration (intervention duration ≤ 6 weeks or intervention duration > 6 weeks).

## 3. Results

### 3.1. Study Selection

The flow of the study identification and selection process is summarized in Figure 1. The initial search identified 2708 potentially relevant articles (PubMed n = 1055, Web of Science n = 989, EBSCO n = 244, EMBASE = 261, Sport-Discuss n = 63, Medline n = 94, Manual Search n = 2). Two manually included studies were retrieved from two reviews [44,45]. After the removal of duplicates, we identified 2047 publication records, consisting of 30 publications with full text. Next, these 30 full-text articles were evaluated for eligibility, and 22 of them were excluded. Finally, after completing the thorough full-text review, eight publications (Table 1) consisting of 170 participants were included in the quantitative synthesis [26,31,32,46,47,48,49,50]. No disagreement was met between the two authors and all the papers included complete data that were needed for the review and meta-analysis.

### 3.2. Characteristics of the Included Studies

#### 3.2.1. Participant Characteristics

The characteristics included were outlined in Table 1. In total, 170 participants were included. The sample size of the included studies ranged from 16 to 27 participants, with participants ranging in age from 23 to 35.4 years. Four studies [26,32,46,50] consisted of men only and the other four studies [31,47,48,49] included both men and women. Participants included in the studies were all physically active participants, including recreationally trained participants, as defined by training in a gym or club in spare time [47,48,49,50] and professionally trained participants, as defined by training in professional or national training institutions and participating in competitions [31,32,51,52]. 

#### 3.2.2. Intervention Characteristics

The physical training protocol in the included studies is CT. Specifically, three studies used the CT of resistance training combined with running, cycling, and swimming; two studies used the CT of resistance training combined with HIIT (high-intensity aerobic interval training); one study reported resistance training combined with cross-fit training (WOD: Workout of the Day); one study reported professional footballer training. Dietary interventions in all studies did not report specific daily dietary details but clearly stated that carbohydrate intake was less than 50 g per day. The daily intake of carbohydrates is a minimum of 7% [46] and a maximum of 17% [48], the intake of fat is a minimum of 43% [46], and a maximum of 75–80% [32]. The control group consisted of high carbohydrate diet [26,31,32], habitual diet [46,47,48,50] and restricted diet [49].

Four of the eight studies implemented intervention duration of six weeks or less, and four studies were greater than six weeks. Specifically, intervention duration in one study was of three weeks [32]; two studies were of 30 days [46,50]; one study of six weeks [50]; two of eight weeks [31,49] and two of 12 weeks [26,47].

### 3.3. Quality Assessment

The results of the quality assessment were shown in Table 2. It was demonstrated that all eight studies were scored between 4 and 6, suggesting the quality of the included studies was moderate. The experimental design of four studies was not randomized, and all the included studies did not implement a blinded trial protocol.

### 3.4. Study Outcomes

Details of the outcomes in each study were summarized in Table 1. The body mass was assessed in seven of the studies [26,31,32,46,47,48,50]. Specifically, four studies used DXA (dual energy X-ray absorptiometry) [26,31,46,48]; two studies used bio-electrical impedance (Inbody770 and Inbody230) [47,50] and one used underwater weighing [49]. The lean mass was assessed in five of the studies. Specifically, three studies used DXA [26,46,48] and two studies used bio-electrical impedance (Inbody770 and Inbody230) [47,50]. The body fat percentage was assessed in five of the studies. Specifically, two studies used DXA [26,48]; two studies used bio-electrical impedance (Inbody770 and Inbody230) [47,50] and one used underwater weighing [49]. The VO_2max_ was assessed in five of the studies using GXT (graded exercise text) [26,31,32,47,50]. Time (or distance) to complete the aerobic tests was assessed in five of the studies [26,32,46,47,48]. Specifically, two studies used 100 km total time and 10 km total Time, respectively [26,32]; one only used Yo-Yo total distance [46]; one used total fatigue time [47]; one used total performance time [48]. 

### 3.5. Meta-Analysis

All the studies in the systematic review were included in the following meta-analyses. We performed subgroup analyses to compare the effects of CT combined with LCHF in different populations (i.e., recreationally trained participants or professionally trained participants) and the effects of intervention duration on those outcomes (Table 3). 

#### 3.5.1. Lean Mass

Four studies showed no significant improvement between CT combined with LCHF compared to the control group [48,49,50,53], but another study showed inconsistent results [47]. 

The results of the meta-analysis showed that the CT combined with LCHF intervention did not negatively affect lean mass in adults compared with the control group. The pooled effect size was trivial (five studies; SMD = −0.08, 95% CI −0.45 to 0.3, *p* = 0.69, Figure 2) and no heterogeneity (I^2^ = 0, *p* = 0.99). The funnel plot (Appendix A) and Egger’s test (t = −0.52, *p* = 0.64) indicated evidence of symmetry.

The sub-group analyses showed that compared to control, CT combined with LCHF had trivial effect sizes in both recreationally (three studies; SMD = −0.11, 95% CI −0.56 to 0.34, *p* = 0.64) and professionally (two studies; SMD = −0.01, 95% CI −0.66 to 0.65, *p* = 0.99) trained participants with no heterogeneity (recreationally: I^2^ = 0, *p* = 0.87; professionally: I^2^ = 0, *p* = 0.91). With regard to the duration of intervention, only trivial effect size was observed for durations both greater (two studies; SMD = 0.01, 95% CI −0.58 to 0.6, *p* = 0.9) and less (three studies; SMD = −0.13, 95% CI −0.61 to 0.35, *p* = 0.59) than six weeks (Figure 3).

#### 3.5.2. Body Fat Percentage

Four of five studies showed a decrease in body fat percentage as induced by CT combined with LCHF as compared to control [26,47,48,49], but the other one did not [50].

The results of the meta-analysis showed that the CT combined with LCHF intervention did not induce significantly greater body fat percentage loss as compared to CT combined with the control group. The pooled effect size was small (five studies; SMD = −0.29, 95% CI −0.68 to 0.1, *p* = 0.14, Appendix A) and with low heterogeneity (I^2^ = 8.4, *p* = 0.359). The funnel plot (Appendix A) and Egger’s test (t = −1.38, *p* = 0.26) indicated evidence of symmetry.

The sub-group analysis showed that compared to control, CT combined with LCHF had trivial effect sizes in both recreationally (four studies; SMD = −0.17, 95% CI −0.57 to 0.23, *p* = 0.41) and professionally (one study; SMD = −0.9, 95% CI −1.83 to 0.03, *p* = 0.06) trained participants with no heterogeneity (recreationally: I^2^ = 0, *p* = 0.5). With regard to the duration of intervention, only a small effect size was observed for durations both greater (four studies; SMD = −0.27, 95% CI −0.7 to 0.15, *p* = 0.2) and less (one study; SMD = −0.33, 95% CI −1.09 to 0.44, *p* = 0.4) than six weeks (Appendix A).

#### 3.5.3. Body Mass

Six of seven studies showed a decrease in body mass as induced by CT combined with LCHF compared to control [26,31,32,46,47,48], but the other did not [50].

The results of the meta-analysis showed that the CT combined with LCHF intervention did not significantly improve the body mass of athletes compared to the control group. The pooled effect size was small (seven studies; SMD = −0.21, 95% CI −0.53 to 0.11, *p* = 0.2, Appendix A) and there was no heterogeneity (I^2^ = 0, *p* = 0.99). The funnel plot (Appendix A) and Egger’s test (t = −0.4, *p* = 0.7) indicated evidence of symmetry.

The sub-group analysis showed that compared to control, CT combined with LCHF had trivial effect sizes in both recreationally (three studies; SMD = −0.17, 95% CI −0.62 to 0.28, *p* = 0.46) and professionally (four studies; SMD = −0.25, 95% CI −0.71 to 0.21, *p* = 0.28) trained participants with no heterogeneity (recreationally: I^2^ = 0, *p* = 0.88; professionally: I^2^ = 0, *p* = 0.97). With regard to the duration of intervention, only a small effect size was observed for durations both greater (four studies; SMD = −0.21, 95% CI −0.63 to 0.21, *p* = 0.33) and less (three studies; SMD = −0.21, 95% CI −0.71 to 0.29, *p* = 0.4) than six weeks (Appendix A).

#### 3.5.4. VO_2max_

Only one of five studies showed significant improvement in VO_2max_ as induced by CT combined with LCHF compared to the control, and the other four did not. 

The results of the meta-analysis showed that the CT combined with LCHF intervention did not significantly improve the VO_2max_ of athletes compared to CT combined with the control group. The pooled effect size was a trivial effect size (five studies; SMD = −0.01, 95% CI −0.4 to 0.37, *p* = 0.95, Figure 4) and no heterogeneity (I^2^ = 0, *p* = 0.92). The funnel plot (Appendix A) and Egger’s test (t = −1.1, *p* = 0.35) indicated evidence of symmetry.

The sub-group analysis showed that compared to control, CT combined with LCHF had trivial effect sizes in recreationally (three studies; SMD = 0.09, 95% CI −0.39 to 0.57, *p* = 0.7) and professionally (two studies; SMD = −0.2, 95% CI −0.83 to 0.43, *p* = 0.54) trained participants with no heterogeneity (recreationally: I^2^ = 0, *p* = 0.52; professionally: I^2^ = 0, *p* = 0.85). With regard to the duration of intervention, trivial effect size was observed for durations both greater (four studies; SMD = 0.04, 95% CI −0.38 to 0.46, *p* = 0.85) and less (one study; SMD = −0.26, 95% CI −1.17 to 0.64, *p* = 0.57) than six weeks (Figure 5).

#### 3.5.5. Time (or Distance) to Complete the Aerobic Tests

Four studies showed no significant improvement between CT combined with LCHF compared to the control group [26,46,47,48], but another study showed inconsistent results [32].

The results of the meta-analysis showed that the CT combined with LCHF intervention did not significantly improve the VO_2max_ of athletes compared to CT combined with the control group. The pooled effect size was a trivial effect size (five studies; SMD = −0.01, 95% CI −0.4 to 0.37, *p* = 0.95, Figure 4) and no heterogeneity (I^2^ = 0, *p* = 0.92). The funnel plot (Appendix A) and Egger’s test (t = −1.1, *p* = 0.35) indicated evidence of symmetry.

The sub-group analysis showed that compared to the control, CT combined with LCHF had trivial effect sizes in recreationally (three studies; SMD = 0.09, 95% CI −0.39 to 0.57, *p* = 0.7) and professionally (two studies; SMD = −0.2, 95% CI −0.83 to 0.43, *p* = 0.54) trained participants with no heterogeneity (recreationally: I^2^ = 0, *p* = 0.52; professionally: I^2^ = 0, *p* = 0.85). With regard to the duration of intervention, a trivial effect size was observed for durations both greater (four studies; SMD = 0.04, 95% CI −0.38 to 0.46, *p* = 0.85) and less (one study; SMD = −0.26, 95% CI −1.17 to 0.64, *p* = 0.57) than six weeks (Figure 5).

## 4. Discussion

To our knowledge, this is the first systematic review and meta-analysis exploring the effects of CT combined with LCHF on body composition and aerobic performance in trained participants. The quality of included studies is moderate. The results suggested that CT combined with LCHF did not significantly affect body composition and aerobic performance in participants. Subgroup analyses also showed that the participant cohort (i.e., recreationally or professionally trained participants) and intervention duration (e.g., six weeks) did not significantly affect the results. More studies with rigorous design and better quality are highly recommended to confirm the observations from this work, which can ultimately help the appropriate design of nutritional programs for LCHF.

Our results showed that CT combined with LCHF did not induce a significant effect on lean mass. Studies showed that the LCHF diet induces nutritional ketosis [54], reducing gluconeogenesis associated with increased muscle protein catabolism when using ketone bodies for energy production, so lean mass is preserved [55]. Meanwhile, we did not observe a significant reduction in body fat percentage and body mass with CT combined with LCHF. One potential reason may be in the included studies, the calorie intake between the diet in the intervention (i.e., LCHF) and that in the control (e.g., restricted diet) may be similar, leading to similar influences on the body composition between groups [56,57]. For example, one study used CT combined with LCHF in the intervention group and CT combined with a calorie-restricted diet in the control group; therefore, the reduction in body fat percentage was not significantly different between the two groups [50]. Future research thus needs to more explicitly and comprehensively characterize the benefits of LCHF on body composition by using an appropriate study design.

We also observed that CT combined with LCHF can help maintain VO_2max_ and aerobic performance, that is, no significant decrease in VO_2max_ and aerobic performance induced by CT with LCHF was observed as compared to CT with other diets. Fat is a major energy source during low to moderate submaximal (<70% VO_2max_) aerobic exercise [58]. Studies have shown that LCHF can increase the rate of fat oxidation, switching from carbohydrates as the primary energy source, thereby improving aerobic performance during prolonged high-intensity aerobic exercise [26,59]. Sustained consumption of a ketogenic diet can enhance maximal fat oxidation to 1.5 g/min at 70% VO_2max_, replacing carbohydrates as the primary energy source [59]. These findings indicate that the use of a ketogenic diet can be considered to improve prolonged aerobic performance [26]. Conversely, exercise economy has been reported to decline when consuming a ketogenic diet [60], with reductions being most prevalent during higher (>70% VO_2max_) exercise intensities [60]. Future studies are needed to explore the effects of CT combined with LCHF on aerobic performance in competition with longer times or greater intensity. Meanwhile, the use of different testing methods or outcomes (e.g., 100 km total time, 10 km total time, Yo-Yo total distance, total fatigue time and, total performance time) of endurance performance may contribute to the variance of findings, which are worthwhile to be implemented in future studies.

Limitations: Several limitations of this work need to be noted. The number of included studies is relatively small, suggesting more studies are needed in this field. The details of the intervention protocol were not clearly provided in some studies [46,47], and in several studies, only the strength component was monitored while the endurance component was overlooked, which may distort the results in this work. A huge variance was observed in the design of included studies, lowering their quality and may potentially affecting the results. Therefore, the findings of this work should be taken with caution. 

## 5. Conclusions

In summary, this comprehensive systematic review and meta-analysis shows that LCHF combined with CT is expected to reduce body fat while maintaining lean mass and aerobic performance, which needs to be confirmed by studies with a more rigorous design. The results of this work provide important knowledge for designing future studies and training practices that implement LCHF combined with CT. 

## Figures and Tables

**Figure 1 ijerph-19-11542-f001:**
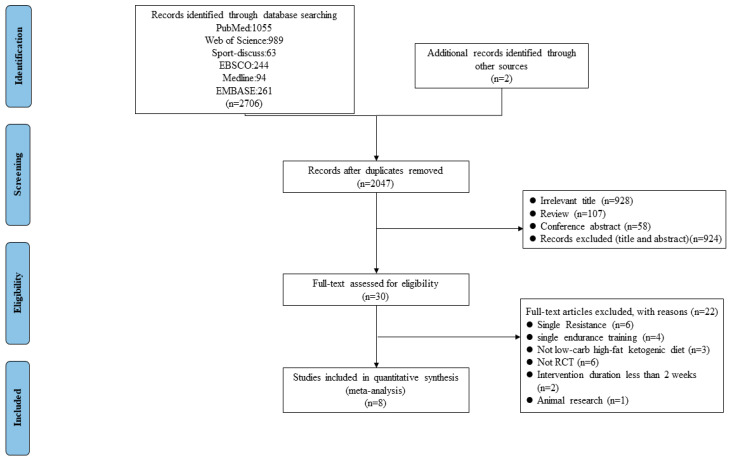
Flow chart for selection of studies.

**Figure 2 ijerph-19-11542-f002:**
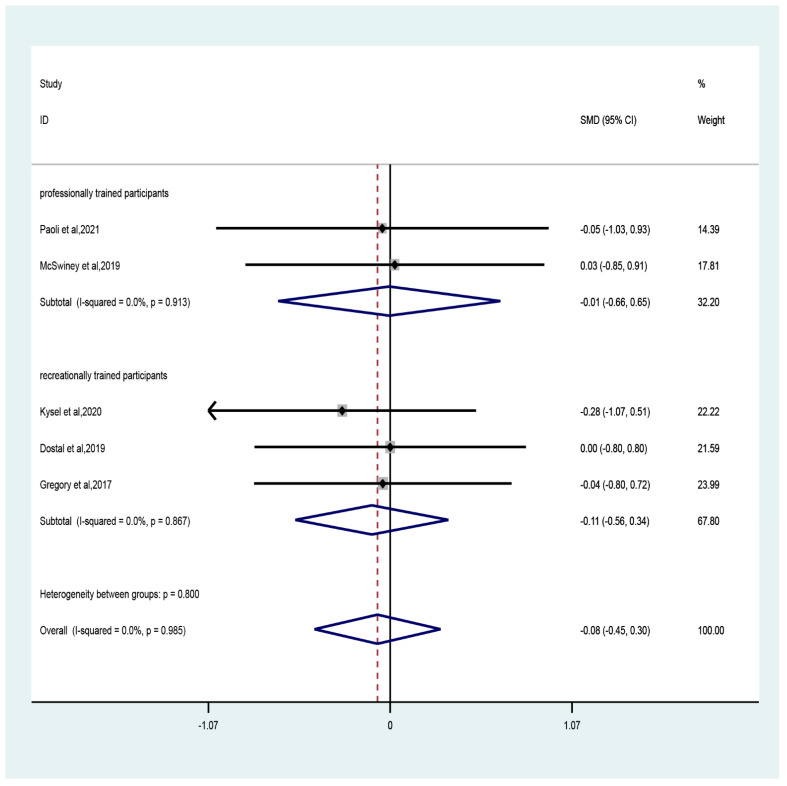
Forest plot of the effect of CT combined with LCHF on lean mass in recreationally trained or professionally trained participants [47,48,49,50,53].

**Figure 3 ijerph-19-11542-f003:**
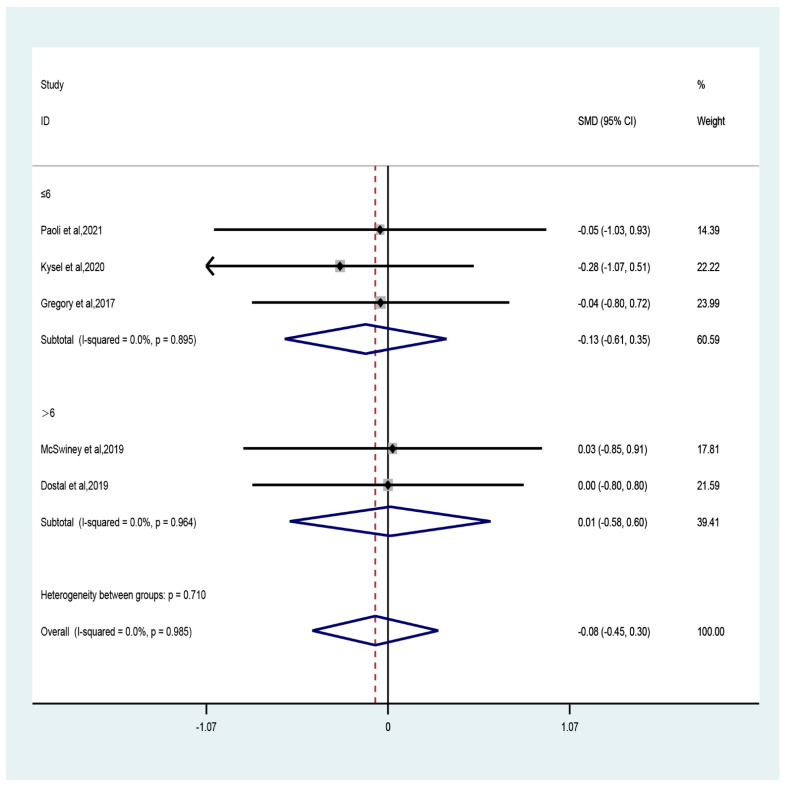
Forest plot of the effect of intervention duration (≤6 weeks or >6 weeks) on lean mass [47,48,49,50,53].

**Figure 4 ijerph-19-11542-f004:**
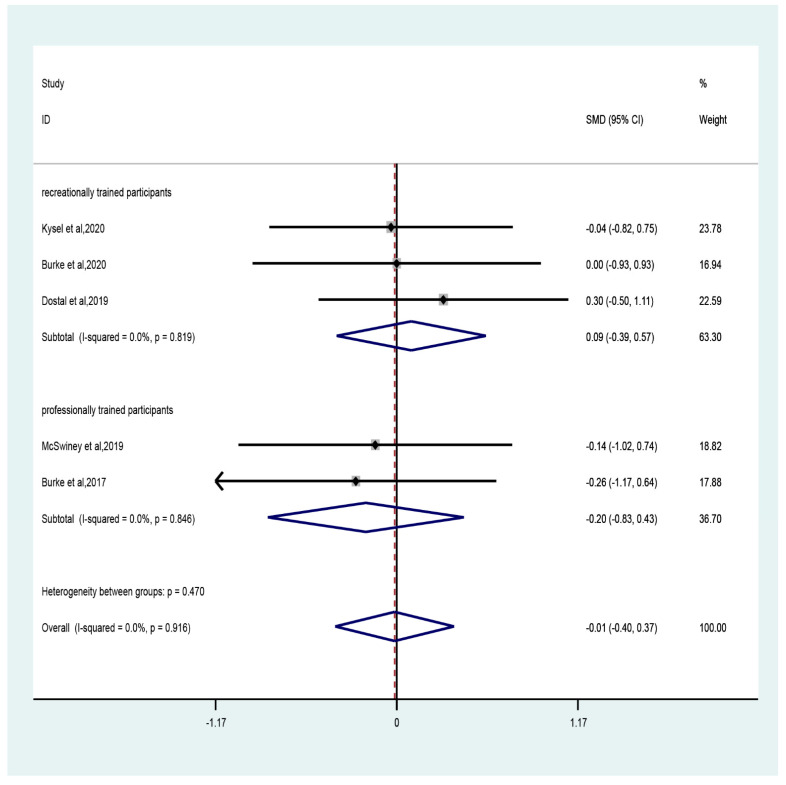
Forest plot of the effect of CT combined with LCHF on VO_2max_ in recreationally trained or professionally trained participants [26,31,32,47,50].

**Figure 5 ijerph-19-11542-f005:**
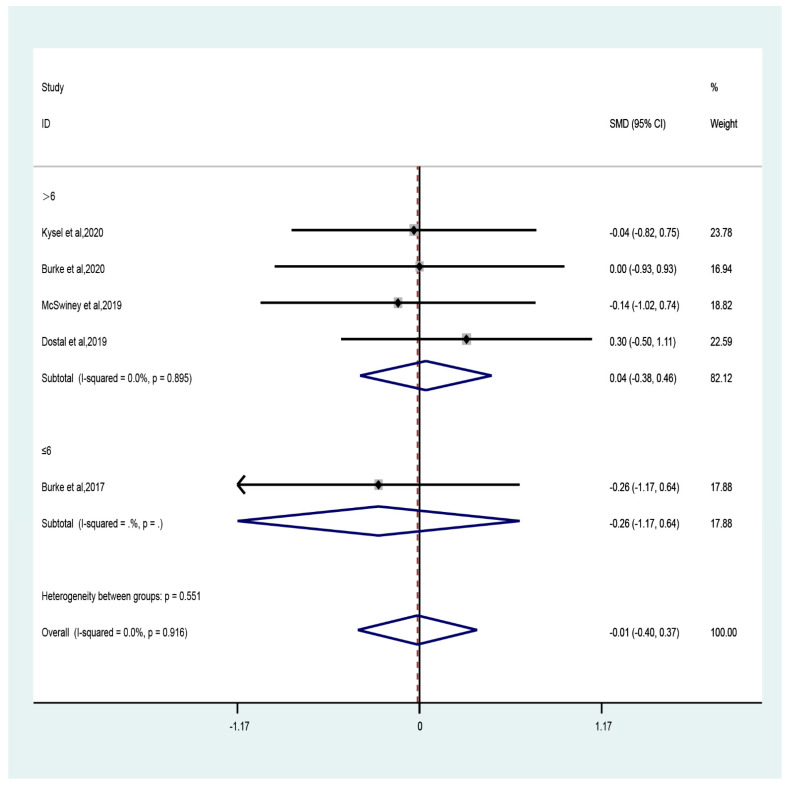
Forest plot of the effect of intervention duration (≤6 weeks or >6 weeks) on VO_2max_ [26,31,32,47,50].

**Table 1 ijerph-19-11542-t001:** Characteristics of the eight included studies.

Study	Participants	Intervention Group-n	Duration Group-n	Intervention	Duration	Macronutrient Ratio in LCHFs	Types of Non-LCHF	Macronutrient Ratio in Non-LCHFs	Ad Libitum (Yes/No)	Outcomes
Meirelles et al., (2019) [49]	at least three months of experience in RT	9	12	RT/AT	8 week	<30 g CHO/d	CONV	CHO: 55% PRT: 15% FAT: 30%	Yes	body mass →; body fat percentage →;
Paoli et al., (2021) [46]	soccer players	8	8	RT/AT	30 day	CHO: 22 ± 5 g PRT: 130 ± 25 g FAT: 132 ± 27 g Total: 1984 ± 430 (Kcal)	WD	CHO: 220 ± 56 g PRT: 129 ± 28 g FAT: 38 ± 10 g Total: 1752 ± 320 (Kcal)	Yes	body weight →; lean soft tissue →; Yo-Yo Total distance →;
Kysel et al., (2020) [50]	experience with RT and AT	13	12	RT/AT	30 day	CHO: 30 g PRT: 1.6 g/kg ≈130 g	RD	CHO: 55% PRT: 15%	Yes	Weight →; muscles →; body fat percentage →; VO_2max_ →.
Burke et al., (2020) [31]	race walkers	10	8	RT/AT	8 week	CHO: 35 ± 3 g PRT: 144 ± 18 g FAT: 326 ± 34 g	HCHO	CHO: 534 ± 77 g PRT: 127 ± 23 g FAT: 69 ± 16 g	Yes	VO_2peak_ →.
McSwiney et al., (2018) [26]	well-trained athletes	9	11	ET/ST/HIIT	12 week	CHO: 41.1 ± 13.3 g PRT: 130.7 ± 35.8 g FAT: 259.3 ± 83.4 g	HCHO	CHO: 400.3 ± 102.7 g PRT: 55.2 ± 10.7 g FAT: 55.2 ± 10.7 g	NO	body mass ↓; body fat percentage ↓; VO_2max_ →; 100 km total time →.
Dostal et al., (2019) [47]	well-trained athletes	12	12	HIIT/RT	12 week	CHO: <50 g	HD	/	NO	body mass (Not counted); skeletal muscle mass (Not counted); body fat percentage (Not counted); total fatigue Time →; VO_2max_ →.
Burke et al., (2017) [32]	race walkers	10	9	RT/AT	3 week	CHO: <50 g PRT: 15–20% FAT: 75–80%	HCHO	CHO: 60–65% PRT: 15–20% FAT: 20%	Yes	VO_2peak_ →; 10 km Total Time ↓.
Gregory et al., (2017) [48]	male and female of all levels of fitness	12	15	RT/WOD	6 week	CHO: 44.42 ± 16.46 g PRT: 91.52 ± 17.34 g FAT: 114.54 ± 25.23 g	WD	CHO: 187.19 ± 68.01 g PRT: 80.45 ± 18.61 g FAT: 73.47 ± 18.86 g	Yes	Weight ↓; Lean mass →; body fat percentage ↓; total Performance time →.

RT: Resistance Training; AT: Aerobic Training; ET: Endurance Training; ST: Strength Training; HIIT: High-Intensity Interval Training; WOD: Workout of the Day; LCHF: Low-Carbohydrate High-Fat Diet; CHO: Carbohydrate; PRT: protein; CONV: Conventional Diets; WD: Western Diet; RD: Restricted Diet; HCHO: High Carbohydrate Diet; HD: Habitual Diet.

**Table 2 ijerph-19-11542-t002:** Quality assessment of included studies (n = 8).

Study	Eligibility Criteria	Random Allocation	Concealed Allocation	Similarity Baseline	Subject Blinding	Therapist Blinding	Assessor Blinding	>85% Retention	Intention-to-Treat	Between-Group Comparisons	Point and Variability Measures	Total Score
Meirelles et al., (2019) [49]	1	1	0	1	0	0	0	0	1	1	1	6
Paoli et al., (2021) [46]	1	1	0	1	0	0	0	0	1	1	1	6
Kysel et al., (2020) [50]	1	1	0	1	0	0	0	0	1	1	1	6
Burke et al., (2020) [31]	1	0	0	0	0	0	0	1	1	1	1	5
McSwiney et al., (2018) [26]	1	0	0	1	0	0	0	1	1	1	1	6
Dostal et al., (2019) [47]	1	0	0	1	0	0	0	1	1	1	1	6
Burke et al., (2017) [32]	1	0	0	0	0	0	0	0	1	1	1	4
Gregory et al., (2017) [48]	1	1	0	1	0	0	0	0	1	1	1	6

1: qualified, 0: unqualified.

**Table 3 ijerph-19-11542-t003:** Overall and subgroup analysis results regarding the effects.

Outcomes	Overall and Subgroup Analysis	No. of Studies	Treatment Effect SMD (95% CI)	*p* Value	Test of Heterogeneity		
					χ^2^	*p* Value	I^2^ (%)
Lean Mass	Overall	5	−0.08(−0.44, 0.3)	0.69	0.36	0.99	0
	Recreational Trained Participants	3	−0.11(−0.56, 0.34)	0.64	0.29	0.87	0
	Professionally Trained Participants	2	−0.01(−0.66, 0.65)	0.99	0.01	0.91	0
	≤6	3	−0.13(−0.6, 0.35)	0.59	0.22	0.9	0
	>6	2	0.01(−0.58, 0.61)	0.98	0	0.96	0
Percentage Body Fat	Overall	5	−0.29(−0.66, 0.08)	0.13	4.37	0.36	8.4
	Recreational Trained Participants	4	−0.17(−0.57, 0.23)	0.41	2.93	0.5	0
	Professionally Trained Participants	1	−0.9(−1.83, 0.03)	0.06	-	-	-
	≤6	1	−0.33(−1.1, 0.44)	0.4	-	-	-
	>6	4	−0.27(−0.7, 0.15)	0.2	4.35	0.23	31.1
Body Mass	Overall	7	−0.21(−0.53, 0.11)	0.2	0.57	0.99	0
	Recreational Trained Participants	3	−0.17(−0.62, 0.28)	0.46	0.26	0.88	0
	Professionally Trained Participants	4	−0.25(−0.72, 0.21)	0.28	0.25	0.97	0
	≤6	3	−0.21(−0.71, 0.29)	0.41	0.05	0.97	0
	>6	4	−0.21(−0.63, 0.21)	0.33	0.52	0.92	0
Time (or Distance) to Complete the Aerobic Tests	Overall	5	−0.02(−0.41, 0.37)	0.1	4.34	0.36	7.8
	Recreational Trained Participants	2	0.13(−0.43, 0.68)	0.66	0.08	0.78	0
	Professionally Trained Participants	3	−0.16(−0.7, 0.38)	0.57	3.74	0.15	46.5
	≤6	3	−0.16(−0.67, 0.35)	0.54	3.43	0.18	41.7
	>6	2	0.17(−0.42, 0.77)	0.57	0.23	0.63	0
VO_2max_	Overall	5	−0.01(−0.4, 0.37)	0.95	0.96	0.92	0
	Recreational Trained Participants	3	0.09(−0.39, 0.57)	0.7	0.4	0.52	0
	Professionally Trained Participants	2	−0.2(−0.83, 0.43)	0.54	0.04	0.85	0
	≤6	1	−0.26(−1.17, 0.64)	0.57	-	-	-
	>6	4	0.04(−0.38, 0.46)	0.85	0.6	0.9	0

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
