# Peer review of "The Effects of Concurrent Training Combined with Low-Carbohydrate High-Fat Ketogenic Diet on Body Composition and Aerobic Performance: A Systematic Review and Meta-Analysis"

_ijerph, 2022, doi:10.3390/ijerph191811542_

Round 1

Reviewer 1 Report

Line:      Comment:

71           Remove the word “here”

73           The word “will” seems too strong, perhaps “may” is better.

85           Change “will be“ to “was”

101         the comma in “used, or” can be removed.

107         Change “when the disagreement on decision happened,” to “when a decision disagreement happened,”

118         Sentence ends with the words “to request”   You need to complete the sentence by stating what was requested. Also, add a period after [36].

147         Remove the word “here”

155-158                Figure 1 should be in Methods section.

Table 1.                Forth column’s title: “Grou” and “p-n”, can it be fixed? In the Macronutrient columns, the numbers without units appear to be in grams, correct?

164         Table 1 indicates that only about half of the 170 participants underwent the CT condition. Am I reading this correctly?

Table 2 has a lot of problems because of too many characters in a line. What is SMD (Treatment effect?)?

334         Perhaps “recommended” instead of “demanded” would be better.

353 to middle of 355. It is not clear what you are trying to indicate here. You indicate that LCHF both increases Fat oxidation and switches to Carbs as a Primary energy source. Instead of “switching to carbohydrates” do you mean “switching from carbohydrates” ? The references for these lines appear to indicate no improvement or a decrement in performance with LCHF.

357 to 358           Reference 16 seems to indicate the opposite of what is stated here.

Author Response

Response to Reviewers and Editor:

We are really grateful for all the help and suggestions from Reviewers and Editor. We believe the helpful comments we received from Reviewers have enabled us to significantly strengthen the scientific merit of our manuscript. Please check the following specific responses. We have highlighted revisions in red in the manuscript.

Response to Reviewer: 1
Comments to the Author

71           Remove the word “here”

Thanks. We have revised it based on your suggestion.

73           The word “will” seems too strong, perhaps “may” is better.

Thank you so much for these very helpful suggestions. We have revised.

85           Change “will be” to “was”

Thanks. We have revised.

101         the comma in “used, or” can be removed.

Thanks. We have revised it based on your suggestion.

107         Change “when the disagreement on decision happened,” to “when a decision disagreement happened,”

Thanks for this very helpful suggestion. We have revised.

118         Sentence ends with the words “to request”. You need to complete the sentence by stating what was requested. Also, add a period after [36].

Thanks. We have revised the incorrect grammar.

147         Remove the word “here”

Thanks. We have revised.

155-158                Figure 1 should be in Methods section.

Thanks. We have revised.

Table 1.                Forth column’s title: “Grou” and “p-n”, can it be fixed? In the Macronutrient columns, the numbers without units appear to be in grams, correct?

Thanks. We have revised it based on your suggestion.

Yes, in grams. We have revised it based on your suggestion.

164         Table 1 indicates that only about half of the 170 participants underwent the CT condition. Am I reading this correctly?

Sorry for the confusion. All participants (n=170) included the intervention (n=83) and control (n=87) groups.

Table 2 has a lot of problems because of too many characters in a line. What is SMD (Treatment effect?)?

Thanks. We have revised it based on your suggestion.

Yes, SMD is treatment effect. We have revised.

334         Perhaps “recommended” instead of “demanded” would be better.

Thanks. We have revised it based on your suggestion.

353 to middle of 355. It is not clear what you are trying to indicate here. You indicate that LCHF both increases Fat oxidation and switches to Carbs as a Primary energy source. Instead of “switching to carbohydrates” do you mean “switching from carbohydrates” ? The references for these lines appear to indicate no improvement or a decrement in performance with LCHF.

Sorry for the confusion. We have revised the wording to make it clear. Please check.

357 to 358           Reference 16 seems to indicate the opposite of what is stated here.

Sorry, it was an operational error, this was not my original reference. We have revised.

Reviewer 2 Report

The work presented to me for review is a well-prepared systematic review and meta-analysis on an important topic, which is the impact of the low-carbohydrate high-fat ketogenic diet in combination with concurrent training on the aerobic performance and body composition. Due to the popularity of this type of diet among people practicing sports (both recreationally and professionally), the topic discussed in this manuscript is very interesting. Only 8 studies with a total of 170 participants met the requirements set by the authors. However, in my opinion, this amount is sufficient to assess the results. The review method was presented correctly and concisely. The inclusion criteria for the review are accurate. Study design and statistical analysis were described and used correctly. Results are clearly presented as figures and tables. The discussion is properly conducted. The authors also presented several limitations of their study – I agree with all factors listed in this section.

Please find my comments below regarding minor corrections to the manuscript that I suggest:

Results

Please clarify the "other sources" which appear in Figure 1. In line 160 you mention "Manual search = 2". Please clarify where these publications were found.

Table 1.: I suggest to change its title to: Eight research studies (...)

Author Response

Response to Reviewers and Editor:

We are really grateful for all the help and suggestions from Reviewers and Editor. We believe the helpful comments we received from Reviewers have enabled us to significantly strengthen the scientific merit of our manuscript. Please check the following specific responses. We have highlighted revisions in red in the manuscript.

Response to Reviewer: 2
Comments to the Author

Please clarify the "other sources" which appear in Figure 1. In line 160 you mention "Manual search = 2". Please clarify where these publications were found.

Thanks. These two studies include: "The Effects of a Ketogenic Diet on Exercise Metabolism and Physical Performance in Off-Road Cyclists" from “Influences of Ketogenic Diet on Body Fat Percentage, Respiratory Exchange Rate, and Total Cholesterol in Athletes: A Systematic Review and Meta-Analysis"; “A Low-Carbohydrate Ketogenic Diet Combined with 6-Weeks of Crossfit Training Improves Body Composition and Performance" from "Effects of resistance training combined with aketogenic diet on body composition: a systematic review and meta-analysis"

Table 1.: I suggest to change its title to: Eight research studies (...)

Thanks. We have revised it based on your suggestion.